# Multivariate Relationships among Carcass Traits and Proximate Composition, Lipid Profile, and Mineral Content of *Longissimus lumborum* of Grass-Fed Male Cattle Produced under Tropical Conditions

**DOI:** 10.3390/foods10061364

**Published:** 2021-06-12

**Authors:** Lilia Arenas de Moreno, Nancy Jerez-Timaure, Nelson Huerta-Leidenz, María Giuffrida-Mendoza, Eugenio Mendoza-Vera, Soján Uzcátegui-Bracho

**Affiliations:** 1Facultad de Agronomía, Instituto de Investigaciones Agronómicas, Universidad del Zulia, Box 15205, Maracaibo, Zulia 4001, Venezuela; lilia.arenas@gmail.com; 2Instituto de Ciencia Animal, Facultad de Ciencias Veterinarias, Universidad Austral de Chile, Valdivia 5090000, Chile; nancy.jerez@uach.cl; 3Department of Animal and Food Sciences, Texas Tech University, Box 42141, Lubbock, TX 79409-2141, USA; 4Facultad de Medicina, Universidad del Zulia, Box 15131, Maracaibo, Zulia 4001, Venezuela; mariagvm@gmail.com; 5Facultad de Ingeniería, Universidad del Zulia, Box 15131, Maracaibo, Zulia 4001, Venezuela; mendozaeb64@gmail.com; 6Facultad de Ciencias Veterinarias, Universidad del Zulia, Box 15131, Maracaibo, Zulia 4001, Venezuela; sojanuzcategui@gmail.com

**Keywords:** *longissimus dorsii lumborum*, multivariate analyses, proximate composition, fatty acid profile, mineral content, carcass traits, tropical beef cattle

## Abstract

Hierarchical cluster (HCA) and canonical correlation (CCA) analyses were employed to explore the multivariate relationships among chemical components (proximate, mineral and lipidic components) of lean beef *longissimus dorsii lumborum* (LDL) and selected carcass traits of cattle fattened on pasture under tropical conditions (bulls, *n* = 60; steers, *n* = 60; from 2.5 to 4.0 years of age, estimated by dentition). The variables backfat thickness (BFT), Ca, Mn, Cu, C14:0, C15:0, and C20:0 showed the highest coefficients of variation. Three clusters were defined by the HCA. Out of all carcass traits, only BFT differed significantly (*p* < 0.001) among clusters. Clusters significantly (*p* < 0.001) differed for total lipids (TLIPIDS), moisture, dry matter (DM), fatty acid composition, cholesterol content, and mineral composition (except for Fe). The variables that define the canonical variate “CARCASS” were BFT and degree of marbling (MARBLING). TLIPIDS was the main variable for the “PROXIMATE” canonical variate, while C16:0 and C18:1*c* had the most relevant contribution to the “LIPIDS” canonical variate. BFT and MARBLING were highly cross-correlated with TLIPIDS which, in turn, was significantly affected by the IM lipid content. Carcass traits were poorly correlated with mineral content. These findings allow for the possibility to develop selection criteria based on BFT and/or marbling to sort carcasses, from grass-fed cattle fattened under tropical conditions, with differing nutritional values. Further analyses are needed to study the effects of sex condition on the associations among carcass traits and lipidic components.

## 1. Introduction

Beef is known as one of the main sources of protein with high biological value, bioavailable minerals (Fe, Zn and P), vitamins of the B-complex (B1, B2, B3 B6 and B12) and other nutritional components (D, E, and β-carotenes) [1,2,3,4,5]. It is also a nutritional source of monounsaturated (MUFA) and essential polyunsaturated (PUFA) fatty acids (omega 3 and omega 6) with dietary and functional properties, and therapeutic effects [6,7,8]. The main benefits of beef consumption for nutrition and health are closely related to its unique chemical composition [2,3,4].

It is well known that intrinsic factors like species, breed, gender, age, and the structure of the type of muscle [3,9,10,11,12,13,14,15], and extrinsic factors such as animal nutrition and pre-slaughter conditions [16,17,18] are largely responsible for the variation found in carcass traits, beef sensory attributes, and nutrient composition. Nutrient composition of grass-fed beef has been a subject of study worldwide [15,19,20,21,22,23] and although there is a general perception that its consumption brings health benefits to consumers, there is no consensus on this matter [24]. Numerous studies on the effect of castration reviewed by Huerta and Ríos [25] have demonstrated that carcasses of castrated males (steers) accumulate more fat than their non-castrated counterparts (bulls); however, the influence of castration on the nutrient composition of lean, grass-fed beef (i.e., fatty acids, cholesterol, and minerals) has been less studied in the tropics, particularly in cattle with *Bos indicus* influence. A couple of reports in Brazil [26,27] indicate that the intramuscular fat (IMF) of bulls contains more PUFA and exhibit a higher PUFA/SFA ratio than steers. These findings are explained by the larger muscle mass and leaner beef of bulls, and therefore, a more abundant content of membrane phospholipids of muscle cells [28]. The comparison of lean meats from grass-fed bulls vs. steers in cholesterol or mineral content has not indicated significant differences [3].

For decades, the meat industry and scientists have used carcass characteristics to predict palatability-related attributes and/or consumer acceptability. Indeed, most of the carcass quality grading systems rely on the relationships between individual (or combined) carcass traits and sensory attributes of meat [25]. Key characteristics that describe the beef carcass include carcass weight, physiological maturity (often using dentition or ossification as a proxy for age), sex, fat cover and colour, and conformation. Depending upon the country, marbling and lean colour and/or texture have often been added as quality traits to refine the carcass evaluation technique [29,30].

To our knowledge, there is limited information regarding the nutritional quality/value of meat specifically focused on a possible relationship between the anatomical or other physical characteristics of the intact beef carcass and its meat nutrient composition. This information gap needs to be addressed/closed particularly for beef produced under tropical, grass feeding conditions given the alleged health claims linked to its consumption [31]. We propose that, to determine any relationship, all of these traits must be simultaneously considered by using a multivariate analysis approach. Jeong et al. [32] investigated the relationships between the content of IMF, the fatty acid composition, and characteristics of the muscle fibre in the longissimus thoracis of pork. These researchers employed the principal components analysis (PCA) and hierarchical cluster analysis (HCA), an appropriate example of the applicability of this type of statistical approach. Similarly, Patel et al. [33] used multivariate analyses to explore the relationship among animal and carcass characteristics, beef (*longissimus thoracis*) quality traits, and lean meat mineral composition (20 elements). In this case, the researchers employed a combination of univariate (simple correlation) and multivariate (factorial analysis) techniques that allowed them to compare the relationship between minerals, not only individually but also in a factorial fashion (five factors) with the animal/carcass performance and the beef quality traits. This study [33] only included the carcass weight as one of the three performance characteristics. Both previous investigations [32,33] indicate the need for studying complex relationships employing a multivariate approach, that may include a large number of variables. In this case the Canonical Correlation analysis (CCA), offers a promising multivariate method to complement other techniques. CCA has been widely used in agricultural science [34,35,36,37] to explore the interrelation between multiple variables, relationships that could be symmetric, that is, without a dependency relationship among them, or asymmetric, when one of the sets is dependent and the other is independent.

The underlying principle of CCA is to investigate the relationship between the variables by developing several independent canonical functions that maximize the correlation between the linear composites known as canonical variates [38]. The CCA represents the bivariate correlation between the two canonical variates in a canonical function. The canonical correlation coefficient measures the strength of association between the variable sets under concern. This technique can assist in the analysis of several traits; furthermore, it may indicate the most relevant factors to the set of variables under study [39,40,41].

Knowing the degree of association of the multivariate relationships between the nutrient composition and the quality traits of dressed beef may allow identifying predictors of the meat nutrient composition that can be assessed on the hanging carcass and, eventually, the possibility to develop selection criteria for sorting carcasses with different nutritional values.

This study aimed to explore the multivariate relationships among chemical components (proximate, mineral and lipid components) of lean beef *longissimus dorsii lumborum* (LDL) and selected carcass traits of cattle fattened on pasture under tropical conditions.

## 2. Materials and Methods

### 2.1. Characteristics of the Sample

Carcass traits and nutrient composition data from a randomly selected group of 120 slaughtered cattle (60 bulls and 60 steers; 2.5 to 4.0 years of age, estimated by dentition) were collected for this observational study. This sample was representative of slaughter male cattle derived from the prevailing production systems in the Venezuelan tropics where livestock is mostly fattened on pasture with little or no supplementation [42]. Out of this group, 9 animals were mixed-breed dairy (predominantly Holstein, Brown Swiss, or dual-purpose cattle without a defined breed predominance) x Zebu breeds; and 110 were mixed-breed cattle with a phenotypic predominance of Zebu breeds.

### 2.2. Harvesting, Carcass Classification and Sample Collection

The animals were harvested at a commercial packing house following the procedures of the Venezuelan Standards of Bioethics and Biosecurity for Research with Living Organism [43], and the Venezuelan Standard for *Postmortem* Inspection of Cattle [44]. After being weighed, carcasses were chilled at 2–4 °C. After 48 h *postmortem*, the chilled carcasses were subjected to evaluation. Skeletal and lean maturity (SM and LM, respectively) scores and subcutaneous backfat thickness (BFT) were determined following USDA guidelines [45]. The subcutaneous fat cover (CFINISH) was evaluated using a four-level scale: 1 = Uniform; 2 = Uneven; 3 = In patches; 4 = Devoid [46]. The degree of marbling (MARBLING) was evaluated according to Decreto Presidencial N° 181, using a descriptive scale: 1 = practically devoid, 2 = traces, 3 = slight and 4 = small amount [47].

After evaluation, chilled carcasses were cut out following conventional butchering procedures according to regulation 792-82 of the Venezuelan Commission for Industrial Standard [48], trimmed to 6.4 mm fat cover, and fabricated into commercial cuts. Muscle samples (2.5 cm thick) from the most anterior (cranial) part of the LDL muscle were excised, individually vacuum packaged, identified by animal number, frozen at −30 °C and stored at −20 °C until the final preparation for the proximate analyses. Samples were partially thawed at 4 °C (to avoid fluids losses), trimmed of visible adipose and connective tissue, and homogenized in a Black & Decker™ food processor. Each homogenized sample was subdivided into smaller portions (subsamples) which were packaged in 50 g-zip-lock bags (4–5 bags) and identified by animal number. Bags containing homogenized subsamples were assigned to each type of chemical analysis (proximate, mineral, or lipid profile analysis) and immediately processed accordingly. The remaining bags were preserved at −20 °C as spare samples in the event that a confirmatory analysis was needed. A flowchart (Appendix A) illustrates sample handling for chemical analyses. All the samples were analyzed in duplicate [49].

### 2.3. Proximate Composition Analysis

Duplicate samples were analyzed for crude protein (CP) content following the Kjeldahl procedure; moisture (WATER) and dry matter (DM) were estimated by weight loss at 105 °C for 24 h, and ash at 550 °C during 6 h [50]. Total lipids (TLIPIDS) content was determined by extracting with a 2:1 chloroform:methanol mixture according to the method of Folch et al. [51] with some modifications as described by Slover & Lanza [52].

### 2.4. Mineral Analysis

Duplicates of 10.0 g of ground meat were calcined in a furnace at 550 °C for 6 h. Sample handling and mineral analyses were conducted according to the methodology described by Giuffrida-Mendoza et al. [1]. Mineral content was expressed in mg.100 g^−1^ of fresh tissue.

### 2.5. Lipid Profile Analysis

Cholesterol content of each steak sample was determined in triplicate, according to the procedure described by Rhee et al. [53].

Fatty acids (FA) were determined by gas chromatography as described by Slover and Lanza [54]. A duplicate of an aliquot of the lipid extract, corresponding to 25 mg of the total lipids of each sample, mixed with the internal standard (Margaric acid, C17:0 methyl ester) was saponified and esterified with BF_3_/CH_3_OH [55] to yield fatty acid methyl esters (FAME). FAME were analyzed following the procedure described by Uzcátegui-Bracho et al. [49].

### 2.6. Data Analysis

The data analysis was performed using the IBM SPSS 23 statistical software [56]. The original, historical data consisted of 120 samples, being reduced to 109 after carrying out preliminary analyses. Univariate analyses were used to evaluate descriptive statistics, kurtosis, skewness, and detection of outliers. Multivariate analyses allowed to detect and treat the possible atypical values and to verify conformity with the basic assumptions of randomness, multivariate normality, and homoscedasticity of the variance. For exploring if any noise was caused for the inclusion of 9 observations (mixed dairy x zebu breed types) the statistical analyses were run again with 100 subjects phenotypically classified as predominantly Zebu crossbreds. The statistical output of this exploratory analysis showed the canonical correlations between the selected carcass traits and the three groups of chemical variables (proximate components, lipid profile, mineral components) were like those found in the previous run with 109 subjects, thus proving that the inclusion of these mixed dairy x Zebu cattle did not cause significant changes in the results. In fact, its inclusion introduced more variability to the sample, which enriched the results.

Two hierarchical cluster analyses (HCA) were performed. The first HCA was applied to explore the presence of any pattern or relationship between the 32 variables under study (except for the categorical variables CFINISH and MARBLING), using the linkage (between groups) method. To measure the degree of association between variables, Pearson’s correlation coefficient was applied with the measurement transformed into absolute values. The second HCA was applied to group all the samples using Ward’s method with the squared Euclidean distance measure and considering the sex condition to describe how the variables are presented within each cluster.

To validate the clusters obtained, an ANOVA with two main factors (sex condition and cluster) was applied on each variable. The results from the two HCA were represented by dendrograms. To analyze the relationship among the subgroups of the variables proximate, mineral, and lipid components with respect to the subgroups of carcass traits, a canonical correlation analysis (CCA) was carried out. Wilk’s Lambda and Bartlett tests were used to determine the significance of canonical correlations.

The acronyms of the variables studied in this research and their definitions are shown in Table 1.

## 3. Results

### 3.1. Descriptive Statistics for Carcass Traits, Proximate Composition, Mineral Content, and Fatty Acid Composition of Beef

The descriptive statistics of the experimental data are presented in Table 2 and Table 3. The variables BFT, Ca, Mn, Cu, C14:0, C15:0, and C20:0 showed the highest coefficients of variation. In general, this sample of grass-fed beef carcasses had relative low values of BFT (0.1–1.2 cm) and MARBLING levels. Frequency and percentage distribution of MARBLING levels (i.e., Practically devoid, Traces and Slight amounts) in the filtered data (*N* = 109) were 48 (44%), 24 (22%) and 37 (33.9%), respectively (values are not presented in tabular form). Among the proximate components, TLIPIDS from separable lean only presented the greatest variation (between 0.93 and 6.67 g.100 g^−1^). Out of the 30 fatty acids under study, the most abundant were C16:0 (0.028–1.288), C18:0 (0.053–0.705), C18:1c (0.27–1.749) and C18:1t (0.117–0.981). Overall, MUFA constituted 57.65% of the total; PUFA represented less than 5% of the total and the rest (37.35%) corresponded to SFA.

### 3.2. Characterization of the Carcass Traits and Chemical Components of Beef Longissimus Lumborum Muscle by HCA

The first HCA allowed to determine how variables were grouped by degree of similarity as calculated by the squared Euclidean distance with a similarity index ranging from 0 (higher similarity) to 25 (lower similarity); a close distance between variables indicating high correlation. This first HCA also allowed to provide a simple representation of the total data composed of 32 variables and to explore how the variables correlated to each other. Figure 1 shows that most of the variables in this dataset, tended to cluster in the same subgroup (carcass traits, proximate composition, mineral content or lipidic composition). The variable CHOLEST was grouped with LM. The variables Fe and Cu were clustered with ash and CP, and the variables TLIPIDS and C16:1 were proximate with the second smallest distance (Figure 1).

The HCA by samples resulted in a dendrogram with three clusters, sufficiently distant to expect relatively different values among the groups (Appendix A). ANOVA and multiple range tests (at 5% of significance) were applied to validate and describe the clusters. Results of these analyses are presented in Table 4 and Table 5. Among the carcass traits, only BFT differs significantly (*p* < 0.001) among clusters (Table 4). In the proximate composition data, TLIPIDS, moisture, and DM resulted different among clusters. With the only exception of Fe, the mineral composition also differed (*p* < 0.0001). Clusters differed (*p* < 0.0001) in fatty acid composition and CHOLEST (Table 5; *p* < 0.05).

Cluster 1 is mainly represented by steers with the highest values in BFT (0.724 cm) and TLIPIDS (4.16 g.100 g^−1^); therefore, this subgroup also exhibited the highest values in most of the fatty acids evaluated (*p* < 0.0001). Cluster 2 is mostly composed of bulls with the lowest BFT (0.226 cm) and total lipids (1.81 mg.100 g^−1^; *p* < 0.05). On the other hand, cluster 3 was more balanced, comprised of 56.5% of bulls and 53.5% of steers, with similar values in BFT and TLIPIDS; this cluster represented the group of samples with similarities in fat-related traits and differences in mineral content. In general, cluster 3 showed the highest values in Ca, Mg, P, K, Zn, Cu and Mn (*p* < 0.0001). Appendix A illustrated the projection means of variables by each cluster.

### 3.3. Relationship among Subgroups of Variables by CCA

The first canonical correlation between U (representing the proximate composition traits) and V (representing the subgroup of carcass traits) was significant (*p* < 0.0001). Canonical redundancy analysis revealed that the first canonical correlation represents 75.96% of the explained variance, which indicates that there is a high degree of association between carcass traits and the proximate composition.

Standardized canonical coefficients or canonical weights of original variables represent their relative contribution to the corresponding canonical variates U (named as “PROXIMATE”) and V (named as “CARCASS”), respectively. Standardized canonical coefficient or canonical weights of TLIPIDS had the greatest contribution to the canonical variate “PROXIMATE” (Table 6). The correlation coefficient (−0.993) also indicated that this variable makes an important contribution to the constitution of this canonical variate.

Table 7 shows the standardized coefficients and correlations coefficients between the original carcass traits and the canonical variate “CARCASS”. The variables MARBLING and BFT were the largest contributors to the formation of this canonical variate (*r* = −0.836 and *r* = −0.855, respectively). On the other hand, the variable CFINISH had a moderate contribution to the canonical variate “CARCASS” (*r* = 0.751). The very low standardized coefficients of SM, LM, and CW indicated an irrelevant contribution of these variables to the canonical variate “CARCASS”.

Canonical cross correlation describes the correlation between variables and the opposite canonical variate. The variable TLIPIDS showed the highest canonical cross-loading with the canonical variate “CARCASS” (Table 8). On the other hand, the variables MARBLING, BFT, and CFINISH showed an important canonical cross correlation with the canonical variate “PROXIMATE”. This result indicates a strong and significant linear correlation between these four variables.

The variance of the canonical variate “PROXIMATE” associated with the variables of its own group represented 30.70% of the total data variation, which could be attributed to the high loading value of the variable TLIPIDS. The cross variance between “PROXIMATE” and “CARCASS” only accounted for 12.10% of the total variance; a low value that is also associated with the contribution of the TLIPIDS variable. The contribution of the canonical variate “CARCASS” was 34.10% of the total variance. The canonical correlation between PROXIMATE and CARCASS subgroups of variables aligns with the results obtained from the linear correlation between the variables related to fatness (Table 9). The correlation coefficients are all significant and moderate to strong, with values very close to those reached by the components of the canonical variates.

The CCA between the subgroups of variables related to lipid composition traits and the subgroup of carcass traits revealed six canonical correlations. The first canonical correlation was highly significant (*p* = 0.002), showing a not very strong correlation coefficient (*r* = 0.629), and represented 59.17% of the explained variability. Table 10 shows the standardized canonical and correlation coefficients between the original variables and their canonical variate “LIPIDS”.

The C16:0 (loading value = −0.796; *r* = −0.912); and C18:1c (loading value = −0.566; *r* = −0.878) showed the highest correlation coefficients with the canonical variety “LIPIDS”. This indicates that Palmitic and Oleic acids were the fatty acids that mostly contributed to the conformation of the canonical variate “LIPIDS”. The variable cholesterol content presented a low canonical weight and, therefore, low correlation with its canonical variety. 

The main fatty acids (C16:0 and C18:1c) with the highest standardized canonical coefficient also exhibited the highest correlations with the carcass trait variables (Table 11). Other fatty acids, like C14:0, C20:0 and C18:1t presented moderate correlations, but with lesser impact given their low standardized coefficients. On the other hand, carcass traits: CFINISH, BFT, and MARBLING presented a not very strong cross correlation with the canonical variate “LIPIDS” (Table 11).

The variance of the canonical variate “LIPIDS” accounted for 38.8% of the total data variation. The cross variance between “LIPIDS” and “CARCASS” only represented 19.10% of the total variance. This low value is potentially related to a low correlation between these two subgroups of variables, and/or the smaller number of CARCASS variables as compared to the lipidic components.

Six possible canonical correlations were obtained between mineral content and carcass traits; however, none of them were statistically significant. Based on the available data and the statistical technique applied, there is insufficient evidence to demonstrate the existence of any relationship between mineral components and carcass traits.

## 4. Discussion

The carcasses evaluated in this study are representative of South American tropical cattle fattened on pasture, which presents more variation in degrees of fatness, carcass finish, and conformation traits than their counterparts that are subjected to more standardized, intense feeding protocols. The results obtained in this study also concur with carcass characteristics [18,57,58] and nutrient values [59,60,61,62] reported in previous studies conducted in Venezuela with samples of *Longissimus lumborum* taken from crossbred cattle varying in age, sex condition, and diet. Also, these values are similar to those reported for the fresh *Longissimus* muscle derived from Bangladeshi beef (zebu type) finished on pasture [63] and other types of tropical cattle [15]. The mineral content found in this study is within the ranges reported for raw meat from tropical cattle subjected to different management practices [3,15,64,65].

The relatively low Ca content in these beef samples is potentially related to the quality of vegetation consumed/grazed by these animals. Pastures and forages constitute the main food sources for fattening cattle in Venezuela. The nutritional value of pastures depends on the amount of nutrients present in these plant species, which are absorbed from the soil; consequently, it will be the characteristics of the soil that defines the development of the plant with respect to the concentration and availability of the mineral elements present.

According to Araujo [66], tropical grasses can hardly supply all the minerals in amounts adequate to the needs of the animals. The factors that affect the mineral content in forages, in addition to the soil, are: the forage species, the age of the pasture, the yield, the management of the pastures, and climate [66]. Low fertility and high acidity stand out among the most important limitations in quality of most soils in Venezuela. Research carried out in Venezuela to study to the state of mineral nutrition in livestock systems [67,68], showed that calcium levels in pastures were generally poor and this deficiency is reflected in the cattle’s animal tissues. A review of the nutritional value of beef produced under tropical conditions [15], also confirms the low concentrations of this mineral.

The relationship between meat sensory traits and physicochemical characteristics has been studied using multivariate analysis techniques in beef [41,69]; however, no available information was found about the relationships of carcass traits with the chemical composition of the derived meat.

The first HCA allowed a simple representation of the total data with 32 variables and to explore how the variables correlated. The second HCA grouped all the observations by similarities in three clusters with a significant variation in BFT, TLIPIDS, mineral and fatty acid composition. It has been demonstrated that sex greatly influences protein and fat deposition in cattle and defines distinct differences in body composition [27,70,71]; nevertheless, our observations suggest that in grass-fed tropical cattle, the expected differences in body and lipid composition between sexes are not as noticeable. In fact, our results suggest that BFT, TPLIPIDS, and fatty acid composition represented the main variables that defined the clusters. Cluster 3 contained a similar proportion of bulls and steers with the lowest values in BFT and total lipids.

It is noteworthy to highlight that this was not a controlled experiment designed to compare sex conditions. Therefore, genetics, management (slaughter weight and age), nutrition and other confounding factors [72] could affect this type of non-controlled comparison. The inclusion of bulls (intact males) and steers (castrates) in the study was just to provide a balanced random sample of these two sex conditions of slaughter male cattle in the country. Since the fat content is the most variable proximate component in beef, we can hypothesize that the low variability of the sample in levels of marbling (and as a result of IMF), is responsible for this outcome. The low variability in IMF is likely due to two additive factors: genetics and plane of nutrition. Both sex conditions had a common genetic background (*Bos indicus*) and regardless of sex, it is known that *Bos indicus*-influenced cattle individuals, exhibits lower levels of marbling when compared to *Bos taurus* biological types. The second factor could be the low energy content of the grass-based diet which did not facilitate a greater expression of the inherent differences between the sex conditions in terms of lipid content. Finally, it should also be considered that only separable lean was used for chemical analyses. Otherwise, if the meat sample had not been devoid of the surrounding subcutaneous and intermuscular fat depots, the differences between the sex conditions would have been more noticeable because the steers would give loins with a more significant amount of fat per 100 g of fresh tissue than bulls. Needless to say, a relatively greater fat content would bring a concomitant reduction in the concentration of other proximate components.

There is a consensus that the use of a CCA as a multivariate approach is appropriate for evaluating the interrelations among meat quality and carcass traits [41,73]. For this study, CCA is suitable because it measures the magnitude of interrelations between sets of multiple variables [74]. The CCA allows for studying the interrelationship among groups of multiple independent variables and determines the magnitude of the relationships that may exist between subgroups. 

We analyzed canonical correlations by paired groups of variables: proximate composition, lipids, and mineral contents with carcass traits. The correlation coefficients between the original variables and the canonical variates allow establishing the weight of each original variable in the conformation of the canonical variate. The main variables that define the “CARCASS” canonical variate were BFT and MARBLING, meaning that these two variables had the highest contribution to the corresponding canonical variate. TLIPIDS was the main contributing variable for the canonical variate “PROXIMATE”, while C16:0 and C18:1*c* had the largest contribution for the “LIPIDS” canonical variate. 

As expected, carcass fatness-related traits (BFT, MARBLING and CFINISH) exhibited the highest cross-correlations with TLIPIDS (Table 7), suggesting that MARBLING is not the only carcass trait that could affect the content of IMF. Canonical weights are important parameters for defining the contribution of original variables to the canonical variates. However, the understanding of canonical loadings and cross-loadings is critical because these values describe the correlation between original and canonical variates. Our findings represent the first evidence of a strong multivariate relationship of quantitative carcass traits with the chemical composition, particularly TLIPIDS, and the fatty acid composition of lean tissue.

In our study, the C16:0 and C18:1*c* components presented the highest correlations with carcass traits. The palmitic acid (C16:0) represents the main product of the *de novo* fatty acids synthesized from carbohydrates and volatile fatty acids of the diet, and it can be elongated to stearic acid (C18:0), and then to arachidic (C20:0) [75]. Also, the HCA revealed a high correlation between C16:1 and TLIPIDS (Figure 1).

According to the available data and the CCA applied, there is not enough evidence to assume that there is an association between mineral component variables and carcass traits. These results validate the findings of the HCA which indicate that carcass traits have a weak correlation with the mineral composition of the meat. The results obtained in this work are comparable with those of Duan et al. [11], who reported weak but significant correlations among beef mineral concentrations and carcass traits. According to Duan et al. [11], Mg concentration was positively correlated (*p* < 0.05) with all carcass traits but negatively correlated with hot carcass weight, while no significant correlation (*p* > 0.05) was detected between contents of Fe or Zn and carcass traits. Garmyn et al. [76] reported significant correlations between Fe, Zn and marbling levels. Castillo et al. [73] reported that the magnitude of the interrelations among protein, fat, and minerals are different between male (castrated and intact males) and female Saanen goats from 5 to 45 kgs live weight. In our study, individual mineral content did not correlate (*p* > 0.05) with any carcass trait. Age-related carcass traits (carcass weight, skeletal and lean maturity) were not correlated (*p* > 0.05) with the proximate, mineral or lipidic compositions.

BFT represents between 10 to 13% of total carcass fat tissue and it is dependent on genetics, nutrition, and finishing systems of ruminants; also, these may be influenced by sex and age, considering that the nutrient dynamics in the ruminant’s body differs between sexes, and these differences become more evident with age [77]. A meta-analysis study by Al-Jammas et al. [78] reported that BFT and USDA yield grade were the variables most highly related to changes in the weight of adipose tissues in the carcass, suggesting that variations in USDA yield grade and BFT may properly explain the differences in meat chemical composition.

The process of IMF deposition depends on many factors such as sex condition, age, and nutrition [72,79,80]. MARBLING, was also significantly and strongly (*r* > 0.5) related to TLIPIDS, despite exhibiting a very low range of scores [between 1 (Practically devoid) and 3 (Slight); Table 2]. Most likely the reason why marbling was barely second to BFT in the r value (Table 9) was due to the aforementioned narrower range of marbling variation present in these carcasses derived from *Bos indicus*-influenced, grass-fed cattle. In fact, BFT data (Table 2) show a higher coefficient of variation (%) than that of marbling (69.2% vs. 46.4%). Clearly, the use of a descriptive scale without subdividing each degree of marbling into 100 subunits like the USDA counterpart contributed to diminish its variation in our analysis. Differences in methods of assessment of MARBLING can affect its correlation with IMF%. For instance, Giaretta et al. [81] found that the IMF was more correlated with the percentage of marbling evaluated by the J-image analysis (*r* = 0.62) than when the USDA scores were used (*r* = 0.56) while Kruk et al. [82] had reported that the Meats Standards Australia (MSA) marbling ratings were poorly associated with IMF% when compared to other scoring systems. The correlations with IMF% ranged from 0.67–0.79 in this study [82]; however, the authors commented that in other Australian studies, the correlation coefficients with the marbling scores of the Australian Authority for the Uniform Specifications of Meat and Livestock (AUS-MEAT) were lower, ranging from 0.32 to 0.57. Another reason for the not very strong correlation detected between MARBLING and IMF in our study might be related to the very nature of these very lean meats (where “Practically devoid” and “Traces” levels of marbling comprised two thirds of the filtered data). Brackebusch et al. [83] reported a strong linear association between IMF (%) and marbling while Kornasla et al. [84] found that marbling percentage was not very strongly correlated with chemical IMF% (*r* = 0.60). In fact, Silva et al. [85] pointed out that the association between marbling and IMF is not very strong because part of the IMF is invisible and it also depends on the size and shape of the marbling specks. Furthermore, differences in methodologies of lipid extraction may explain the discrepancies among values reported for correlations between IMF% and marbling levels. According to Siebert et al. [86] when meat is low in fat, significantly more total lipids are extracted with polar solvent mixtures (e.g., chloroform:methanol) due to the phospholipid content of tissue membranes. Therefore, it can be speculated that the total lipid extract of these very lean meats having a much higher component of invisible membrane lipids (e.g., phospholipids and lipoproteins) makes it difficult to achieve the stronger correlations (with the marbling scores) found in highly-marbled carcasses.

Rhee et al. [53] reported the relationships of MARBLING (with eight levels from “Moderately Abundant” to “Practically Devoid”) to CHOLEST of beef *longissimus* muscle, and showed that only raw steaks with “Practically Devoid” MARBLING level contained significantly less cholesterol (on wet basis) than did raw steaks with any of the other seven MARBLING scores. In the present study, CHOLEST showed a low loading value, indicating a weak association with carcass traits. Catillo et al. [87] reported a close relationship between pork carcass leanness as defined by the EUROP classification system, and the fatty acid composition of backfat. These authors [87] found that as the lean meat content of the carcass decreased from class E to class O, the backfat total content of SFA increased by more than 4 percent, while the total PUFA content decreased about the same percentage; however, we could not find similar reports in beef. Elucidation of the multivariate and quantitative relationship between BFT, MARBLING and fatty acid composition may be useful for a better understanding of the roles of fat deposition on the nutritional composition of beef produced under the conditions described herein. 

## 5. Conclusions

Canonical correlation analysis is an optimized multivariate technique for evaluating the existence or non-existence of relationships between complex groups of variables. In this study it proves to be a powerful tool to study the relationship between the selected set of carcass traits and the proximate, lipid and mineral components, particularly when it is expected that certain degree of interaction exists among these three groups of chemical variables. This work demonstrates an important relationship between backfat thickness, marbling and the content of total lipids and fatty acids in beef LDL muscle from cattle fed on pastures under tropical conditions. Instead, carcass traits are poorly associated with beef mineral content. These findings allow for the possibility to develop selection criteria based on BFT and/or marbling to sort carcasses with differing nutritional values. For the experience gained in beef carcass grading in Venezuela, the evaluation of marbling levels requires more intense training and supervision of graders than the BFT measurement. Moreover, marbling is seldom used to grade beef carcasses in the developing, tropical countries. Therefore, it is more practical to use BFT in future regression analyses to explain the variation in lipid composition of beef *longissimus* muscle.

Further analyses are needed to determine the potential influence of sex condition on the magnitude of the associations among carcass traits and beef fatty acid composition.

## Figures and Tables

**Figure 1 foods-10-01364-f001:**
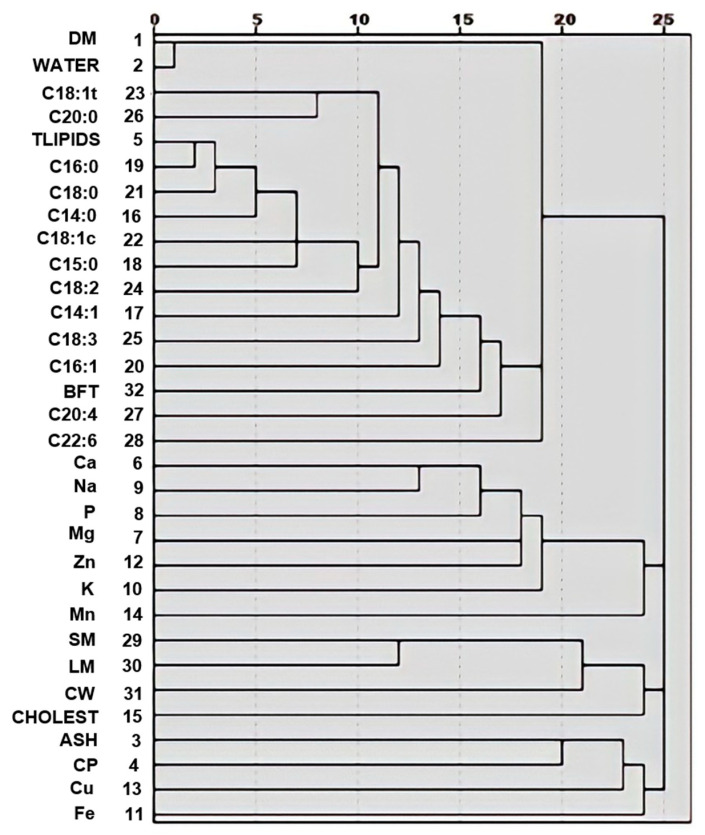
Dendrogram for variables from the hierarchical cluster analysis. Description of acronyms is presented in Table 1.

**Table 1 foods-10-01364-t001:** Acronyms of the variables studied and their definitions.

Abbreviation	Definition
SM	Skeletal maturity
LM	Lean maturity
CW	Carcass weight
BFT	Back fat thickness
CFINISH	Carcass finish
MARBLING	Degree of marbling
TLIPIDS	Total lipids content
DM	Dry matter content
CP	Crude protein content
CHOLEST	Cholesterol content
C14:0	Myristic acid
C14:1	Myristoleic acid
C15:0	Pentadecilic acid
C16:0	Palmitic acid
C16:1	Palmitoleic acid
C18:0	Stearic acid
C18:1*c*	Oleic acid
C18:1*t*	Elaidic acid
C18:2	Linoleic acid
C18:3	α-Linolenic acid
C20:0	Arachidic acid
C20:4ω6	Arachidonic acid
C22:6ω3	Docosahexaenoic acid
SFA	Sum of saturated fatty acids
UFA	Sum of unsaturated fatty acids
MUFA	Sum of monounsaturated fatty acids
PUFA	Sum of polyunsaturated fatty acids
HCA	Hierarchical cluster analysis
CCA	Canonical correlation analysis
“CARCASS”	Canonical variate of the variable of carcass traits
“PROXIMATE”	Canonical variate of the variables of proximate composition
“LIPIDS”	Canonical variate of the variables of lipidic composition
“MINERAL”	Canonical variate of the variables of mineral content
SEXC	Sex condition

**Table 2 foods-10-01364-t002:** Descriptive statistics for carcass traits proximal, and proximate and mineral contents in beef *longissimus lumborum* muscle.

Variable		Mean	SD	Minimum	Maximum	CV
Carcass traits	SM ^1^	212.8	37.96	150.00	350.00	0.179
	LM ^1^	193.1	23.46	150.00	260.00	0.121
	CW, kg	279.5	34.41	207.00	380.00	0.123
	BFT, cm	0.41	0.28	0.10	1.20	0.692
	MARBLING ^2^	1.90	0.88	1	3	0.464
	CFINISH ^3^	2.11	0.69	1	3	0.325
Proximate g.100 g^−1^	DM	26.02	1.29	23.19	29.64	0.05
	Moisture	73.99	1.32	70.36	77.39	0.02
	Ash	1.05	0.15	0.70	1.43	0.14
	CP	20.79	1.53	16.90	24.00	0.07
	TLIPIDS	2.79	1.09	0.93	6.67	0.39
Mineral content mg.100 g^−1^	Ca	2.83	1.58	1.00	8.27	0.560
	Mg	21.73	3.05	14.34	29.27	0.140
	P	210.05	34.68	100.13	322.53	0.165
	Na	82.69	19.97	41.03	119.00	0.242
	K	241.73	59.56	119.78	395.87	0.246
	Fe	1.87	0.49	0.44	3.76	0.265
	Zn	4.14	0.78	2.79	6.60	0.189
	Cu	0.086	0.04	0.024	0.19	0.457
	Mn	0.026	0.014	0.008	0.08	0.533

^1^ Carcasses within the 100–199 maturity range score represent the youngest group (100 is equal to A00 and 199 is equal to A99) ; 200–299: represent carcasses with intermediate, more advanced maturity (200 is equal to B00 and 299 is equal to B99). ^2^ 1 = Practically devoid, 2 = Traces, 3 = Slight, 4 = Small; ^3^ 1 = Uniform; 2 = Uneven; 3 = In patches; 4 = Devoid. Description of acronyms is presented in Table 1.

**Table 3 foods-10-01364-t003:** Descriptive statistics for cholesterol content and fatty acid profile in beef *longissimus lumborum* muscle.

Variable		Mean	SD	Minimum	Maximum	CV
Lipid profile mg.100 g^−1^	CHOLEST	64.96	13.86	30.16	97.34	0.213
	C14:0	0.068	0.033	0.018	0.156	0.481
	C14:1	0.038	0.022	0.002	0.110	0.596
	C15:0	0.079	0.044	0.004	0.223	0.560
	C16:0	0.534	0.245	0.028	1.288	0.460
	C16:1	0.092	0.042	0.023	0.249	0.456
	C18:0	0.285	0.125	0.053	0.705	0.440
	C18:1*c*	0.876	0.320	0.274	1.749	0.365
	C18:1*t*	0.489	0.202	0.117	0.981	0.413
	C18:2	0.076	0.034	0.010	0.163	0.443
	C18:3	0.006	0.003	0.001	0.016	0.453
	C20:0	0.005	0.003	0.001	0.015	0.554
	C20:4ω6	0.013	0.006	0.003	0.034	0.447
	C22: 6ω3	0.025	0.011	0.002	0.062	0.442
	SFA	0.973	0.429	0.275	2.432	0.442
	UFA	1.632	0.563	0.544	3.032	0.345
	MUFA	1.502	0.525	0.488	2.827	0.349
	PUFA	0.121	0.044	0.036	0.234	0.362
	*Cis*	1.156	0.423	0.368	2.637	0.366
	*Trans*	0.491	0.202	0.117	0.981	0.411
	UFA/SFA	1.794	0.552	0.773	4.583	0.308
	MUFA/SFA	1.646	0.488	0.673	4.075	0.298
	PUFA/SFA	0.136	0.057	0.051	0.508	0.418
	*Cis/Trans*	2.528	0.790	1.229	5.709	0.312

Description of acronyms is presented in Table 1.

**Table 4 foods-10-01364-t004:** Comparison of carcass traits, proximate composition, and mineral content of beef *longissimus lumborum* muscle between sex conditions and the three clusters.

Variable	Cluster 1		Cluster 2		Cluster 3		*p* Value	
	Steer (*n* = 25)	Bull (*n* = 3)	Steer (*n* = 7)	Bull (*n* = 26)	Steer (*n* = 22)	Bull (*n* = 26)	Cluster	SEXC
Carcass Traits								
SM ^1^	213.60	193.33	228.57	201.92	211.82	220.38	0.581	0.527
LM ^1^	198.40	173.33	195.71	190.00	195.94	191.56	0.588	0.108
BFT ^2^	0.724	0.333	0.357	0.226	0.510	0.215	<0.0001	<0.0001
CW ^3^	289.48	301.33	247.14	276.42	280.00	278.96	0.066	0.863
Proximate composition (g.100 g^−1^)								
DM	26.78	25.95	25.27	24.97	26.55	26.10	<0.0001	<0.0001
Moisture	73.20	74.05	74.69	75.07	73.47	73.91	<0.0001	<0.0001
Ash	1.04	1.03	1.05	1.08	1.02	1.06	0.669	0.276
CP	20.17	20.50	21.17	21.09	20.83	20.97	0.052	<0.0001
TLIPIDS	4.16	3.59	2.24	1.81	2.83	2.45	<0.0001	<0.0001
Mineral content (mg.100 g^−1^)								
Ca	1.88	2.09	2.58	1.98	4.16	3.59	<0.0001	0.618
Mg	20.26	22.58	18.29	20.47	23.08	24.09	<0.0001	0.049
P	198.27	226.87	201.72	189.04	223.45	231.33	<0.0001	0.075
Na	88.32	96.42	87.70	93.54	72.87	71.79	<0.0001	0.703
K	214.84	215.53	208.57	213.45	268.79	284.93	<0.0001	0.322
Fe	1.98	1.93	1.78	1.83	1.87	1.83	0.445	0.446
Zn	4.04	3.69	3.78	3.75	4.49	4.49	<0.0001	0.541
Cu	0.067	0.064	0.093	0.086	0.109	0.082	0.010	0.535
Mn	0.021	0.024	0.017	0.022	0.026	0.036	<0.0001	0.015

Description of acronyms is presented in Table 1. ^1^ Carcasses within the 100–199 maturity range score represent the youngest group (100 is equal to A00 and 199 is equal to A99) ; 200–299: represent carcasses with intermediate, more advanced maturity (200 is equal to B00 and 299 is equal to B99). ^2^ expressed in cm. ^3^ expressed in kg.

**Table 5 foods-10-01364-t005:** Comparison of lipid profile of beef *l**ongissimus lumborum* muscle between sex conditions within the clusters.

Variable ^1^	Cluster 1		Cluster 2		Cluster 3		*p* Value	
	Steer (*n* = 25)	Bull (*n* = 3)	Steer (*n* = 7)	Bull (*n* = 26)	Steer (*n* = 22)	Bull (*n* = 26)	Cluster	SEXC
Lipid profile (mg.100 g^−1^)								
CHOLEST	65.99	64.88	62.45	63.27	61.244	69.49	<0.0001	0.311
C14:0	0.103	0.092	0.051	0.043	0.068	0.059	<0.0001	<0.0001
C14:1	0.060	0.045	0.025	0.026	0.036	0.030	<0.0001	<0.0001
C15:0	0.131	0.104	0.064	0.046	0.078	0.060	<0.0001	<0.0001
C16:0	0.8280	0.715	0.434	0.311	0.564	0.456	<0.0001	<0.0001
C16:1	0.131	0.117	0.071	0.085	0.081	0.075	<0.0001	0.002
C18:0	0.425	0.402	0.218	0.191	0.283	0.248	<0.0001	<0.0001
C18:1c	1.216	1.209	0.724	0.611	0.917	0.781	<0.0001	<0.0001
C18:1t	0.713	0.601	0.363	0.311	0.493	0.464	<0.0001	<0.0001
C18:2	0.108	0.097	0.056	0.051	0.071	0.075	<0.0001	0.001
α-C18:3	0.008	0.007	0.005	0.004	0.006	0.005	<0.0001	<0.0001
C20:0	0.007	0.005	0.002	0.003	0.005	0.005	<0.0001	0.010
C20:6ω6	0.016	0.011	0.011	0.010	0.013	0.013	<0.0001	0.023
C22:6ω3	0.032	0.024	0.025	0.020	0.025	0.021	<0.0001	<0.0001

^1^ Description of acronyms is presented in Table 1.

**Table 6 foods-10-01364-t006:** Standardized canonical coefficient and canonical correlation coefficient between the original variables and its canonical variate “PROXIMATE”.

Original Variables	Standardized Canonical Coefficient	Canonical Correlation Coefficient
DM	0.230	−0.502
Moisture	0.283	0.510
Ash	0.088	0.071
CP	0.034	0.173
TLIPIDS	−0.966	−0.993

Description of acronyms is presented in Table 1.

**Table 7 foods-10-01364-t007:** Standardized canonical coefficient (canonical weights), and canonical correlation coefficient between the original variables and the canonical variate “CARCASS”.

Original Variables	Standardized Canonical Coefficient	Canonical Correlation Coefficient
SM	0.143	−0.023
LM	−0.037	−0.099
CW	0.027	−0.196
CFINISH	0.297	0.751
BFT	−0.424	−0.855
MARBLING	−0.501	−0.836

Description of acronyms is presented in Table 1.

**Table 8 foods-10-01364-t008:** Cross correlations (canonical cross loadings) between the variables and the opposite canonical variate.

Original Variables	Canonical Variate “CARCASS”	Original Variables	Canonical Variate “PROXIMATE”
DM	−0.316	SM	−0.014
Moisture	0.320	LM	−0.062
Ash	0.045	CW	−0.123
CP	0.109	CFINISH	0.472
TLIPIDS	−0.624	BFT	−0.538
		MARBLING	−0.526

Description of acronyms is presented in Table 1.

**Table 9 foods-10-01364-t009:** Pearson correlation coefficients among carcass fatness-related variables.

Variables	TLIPIDS	CFINISH	BFT	MARBLING
TLIPIDS	1	−0.471 **	0.5532 **	0.519 **
CFINISH	−0.471 **	1	−0.580 **	−0.432 **
BFT	0.532 **	−0.580 **	1	0.549 **
MARBLING	0.519 **	−0.432 **	0.549 **	1

** Significant correlation at *p* <0.01 (bilateral). Number of observations = 109. Description of acronyms is presented in Table 1.

**Table 10 foods-10-01364-t010:** Standardized canonical coefficient (canonical weights), and canonical correlation coefficient between the original variables and their canonical variate “LIPIDS”.

Original Variable	Standardized Coefficient	Correlation Coefficient
CHOLEST	0.082	0.064
C14:0	0.125	−0.738
C14:1	0.065	−0.517
C15:0	−0.030	−0.742
C16:0	−0.796	−0.912
C16:1	0.127	−0.501
C18:0	−0.032	−0.798
C18:1*c*	−0.566	−0.878
C18:1*t*	0.144	−0.679
C18.2	0.257	−0.549
α-C18:3	−0.073	−0.542
C20:0	−0.072	−0.474
C20:4ω6	−0.226	−0.423
C22:6ω3	0.047	−0.344

Description of acronyms is presented in Table 1.

**Table 11 foods-10-01364-t011:** Cross correlations (canonical cross loadings) between the original variables and “CARCASS” and “LIPIDS” canonical variates.

OriginalVariables	“CARCASS”	Variables	“LIPIDS”
CHOLEST	0.045	SM	0.026
C14:0	−0.518	LM	−0.036
C14:1	−0.363	CW	−0.177
C15:0	−0.521	CFINISH	0.603
C16:0	−0.640	BFT	−0.567
C16:1	−0.352	MARBLING	−0.534
C18:0	−0.560		
C18:1c	−0.617		
C18:1t	−0.477		
C18:2	−0.385		
α-C18:3	−0.380		
C20:0	−0.332		
C20:4ω6	−0.297		
C22:6ω3	−0.241		

Description of acronyms is presented in Table 1.

## Data Availability

Data are not available in public datasets, please contact the authors.

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
