# Peer review of "Multivariate Relationships among Carcass Traits and Proximate Composition, Lipid Profile, and Mineral Content of Longissimus lumborum of Grass-Fed Male Cattle Produced under Tropical Conditions"

_foods, 2021, doi:10.3390/foods10061364_

Round 1
Reviewer 1 Report
The overall goal of the study was to determine the multivariate relationship among proximate composition, lipid profile, mineral content, and carcass quality traits. The researchers used various statistical tools such as Hierarchical clustering and Canonical correlation to understand the relationship between variables. This reviewer is not sure about the novelty or practical implication of understanding the relationship between variables. In this omics era, researchers are identifying the molecular level relationship between proteins, genes, metabolites, and related omics to predict variables. These types of analysis can have theoretical value to identify relationships (or probably ignorance of this reviewer). For example, one of the major findings was backfat thickness of animal and lipid content. One can assume if there is a greater deposition of fat tissue, that will be implicated in the entire body.
The experiment is designed well, the sample size is appropriate, and the statistical analysis is ok. There are no major flaws in the manuscript.
Minor comments
Line 47: define AGPI
Line 67: Please rewrite to “limited available information.”
Line 91: Please indicate the full muscle name.
Line 108: Skeletal-
Line 135: Please change to lipid profile. Not sure lipidic is a standard use.
Author Response
Replies to Reviewer 1 are presented in the attached file.

Reviewer 2 Report
COMMENTS FOODS 1231610
Multivariate relationship among proximate composition, lipid profile, mineral content and carcass quality traits of grass-fed beef produced under tropical conditions
WORK’S STRENGTHS
The work points out a statistical method to predict important meat quality parameters, mainly related to nutritional quality, from variables more easily measured such as carcass traits.
From that point of view, the work is interesting since it is focused on recording information through non-destructive methods and providing nutritional information, which is traditionally obtained through analytical and destructive chemical analysis.
The paper is well written and presents a good review of the relevant literature.
However, a few points should be addressed. See specific comments below.
SPECIFIC COMMENTS
Highlights
The highlights reflected properly the main aspects of the manuscript.
Abstract
The abstract is concise and understandable itself. However, BFT abbreviation should be defined in their first mention (line 25), not after.
Introduction
The introduction section is well documented and properly organised.
Materials and Methods
2.2 Harvesting, carcass classification and sample collections
Lines 117-122:
- I have some doubts about the sampling procedure. According to authors, samples were vacuum packaged and frozen. I don’t understand why the partially thawed samples are trimming and minced and refreezing again. In minced meat the thawed losses are much higher than in intact meat. This fact could be altering proximate meat composition, overall in samples as different such as used in the study.
- How many time have samples remained stored under freezing? How were the minced samples thawed?
In order to clarify the sampling procedure, more detailed information about these aspects should be added.
Results
- The approach analysis studied to detect relationship between measures in intact carcasses and meat composition is valuable, but considering the variability of samples, it seems that such feasibility due the wide differences in fat content and fatty acid composition of carcasses. I have doubt about the results in closer animal composition.
- Line 214: (Table 5; P<0.05) should be in the following sentence, since it refers to lipid profile.
Discussion
- Line 343: Give a hypothesis why Ca content are different those reported in USDA, since the rest of meat composition are according to these database
- Line 355-359: Give a hypothesis why the sex was not as related with proximate composition as expected.
- Line 418.The sentence should be rewritten because marbling score is based in standards are well defined and therefore is not so subjective as is indicated.
Conclusion
The authors should punctual the suitability of intact carcass analysis to be the selection criteria for cattle since it may be could not use in general.
Author Response
Replies to Reviewer 2 are presented in the attached file

Reviewer 3 Report
Specific comments:
L25: define BFT at first use; move from line 27
L47: AGPI doesn’t make sense as an abbreviation (unless it’s a translational variation). However, it’s not used again, so there’s really no point in creating an abbreviation for a single use. Delete AGPI.
L53: reference #19 appears to cover intrinsic rather than extrinsic factors and should be moved/renumbered
L54: reference #20 is a study of lamb, but you are using it in reference to beef. This reference is not appropriate and should be omitted.
L99: were the 10 mixed dairy animals evenly split between bulls and steers or was there an uneven distribution. I question the inclusion of these animals in your dataset and would like to know how results would change with their removal. You don’t appear to bring them into the discussion, mainly focusing on zebu (bos indicus-type) cattle.
L117: where specifically was the muscle excised? There could/should be compositional variation if the muscle was removed from the 12th vs 5th rib, so the anatomical point must be included here. Also you compare the compositional results to the small end of the rib, so I assume these came from the 12th rib, but this needs to be explicitly stated here.
L146: please describe why 11 samples were removed. What criteria were used to determine if samples were suitable to remain in the dataset?
L335: were your samples equivalent to the rib, small end or were they derived from a different anatomical location? See comment for L117.
L335-337: your marbling score was representative of Standard (PD to Tr marbling) not Select. So I don’t see why you’re making a point about lipid being 2.4 times lower than what was expected. You’re not comparing apples to apples here. It should have been lower the published values for USDA Select rib. Please revise discussion here.
L376-377: How was marbling not the principal carcass trait affecting IM? Marbling by definition is IM fat. I know later on you suggest that the low range in marbling could contribute to this anomaly, but more discussion is warranted here. To some extent, this makes me question the validity of the marbling scores is they don’t have the strongest relationship to IMF.
L487: should be gender instead of ender
L560-561: the standards have been revised. 2017 is the current version.
Author Response
We did not get a third revision
Round 2
Reviewer 3 Report
No justification was provided for the inclusion of the 10 outlier carcasses (dairy cross). Nor do the authors appear to address this discrepancy in the discussion.
So again - I only said major because there are 10 animals that don't fit the dataset and I would like to know how/if the results change with their removal. At a minimum, better justification is needed for the inclusion of those 10. The authors did not seem to address this at all (at least that I can find).
line 505-509 - in the last version, the authors indicated longissimus thoracis was obtained and have now changed to longissimus lumborum. This is a completely different subprimal and typically has different composition, despite being the same muscle. The authors are still, however, trying to compare the composition of the samples to rib samples (longissimus thoracis) in the USDA database for a higher quality grade (Select as opposed to Standard). This discussion is not productive as it's not the same muscle section or marbling. Revise.
Author Response
Reply to Reviewer No. 3 is attached. Please see the attachment.
